# Duration of Low Temperature Exposure Affects Egg Hatching of the Colorado Potato Beetle and Emergence of Overwintering Adults

**DOI:** 10.3390/insects12070609

**Published:** 2021-07-05

**Authors:** Jianghua Liao, Juan Liu, Zhijian Guan, Chao Li

**Affiliations:** 1Key Laboratory of the Pest Monitoring and Safety Control on Crop and Forest in Universities of Xinjiang Uygur Autonomous Region, College of Agronomy, Xinjiang Agricultural University, Urumqi 830052, China; ljh111196@163.com (J.L.); lj961527860@163.com (J.L.); 2Qapqal County Agricultural Extension Station, Ili 835300, China; g18139214113@163.com

**Keywords:** *Leptinotarsa decemlineata*, cold spell in later spring, developmental duration, survival

## Abstract

**Simple Summary:**

The Colorado potato beetle is a pest of Solanaceae in China, more than 30 species of Solanaceae as host plants of the Colorado potato beetle, and this beetle can cause yield losses to potato farms. Temperature is one of the main factors that affect the growth and survival of insects. In recent years, due to global climate change, sudden drops in short-term temperatures are also frequent, which will affect insects. This paper studies the influence of short-term low temperature on the Colorado potato beetle. In laboratory experiments, we evaluated the effect of short-term low temperature on the Colorado potato beetle eggs, and the emerged date, the number of Colorado potato beetles, were analyzed in combination with the low temperature conditions. Our results show that low temperature had adverse effects on the development of Colorado potato beetle eggs and the emerged of adults Colorado potato beetles, and the longer the low temperature treatment time, the greater the impact.

**Abstract:**

The Colorado potato beetle is a serious pest of Solanaceae in China. In early summer, cold spells in later spring may occur for brief periods in the field environmental conditions, and temperatures often deviate far below the normal temperature for short periods, such as sudden short-term low temperature, may affect the development of Colorado potato beetle eggs. This paper studies the effects of low temperature stress at 8 °C for 0 d, 1 d, 3 d, 5 d, 7 d, and 10 d on the development of Colorado potato beetle eggs. Our results show that egg survival is significantly affected by short-term low temperature exposure. The percentage of eggs hatched is significantly affected by different treatment times (*p* = 0.000)—the percentage of eggs hatched decreases with increased treatment time, and Colorado potato beetles will extend the wintering time of their soil to resist the effects of lower temperatures. Thus, exposure of Colorado potato beetles to a short-term low temperature affects their emergence and population growth; this study could provide information for the occurrence, monitoring, and early warning of Colorado potato beetle during short-term temperature.

## 1. Introduction

The Colorado potato beetle (*Leptinotarsa decemlineata*) belongs to Coleoptera, Chrysomelidae, is an internationally recognized quarantine pest, which can cause significant damage to the potato industry [1,2]. Both the larvae and adults feed on the leaves of the Solanaceae family, young larvae biting out small holes in the leaf and older larvae and beetles also gnawing at the edges of the leaves, when there are no leaves, larvae, and beetles of the Colorado potato beetles gnaw at the offshoots and stems. As Colorado potato beetles damage plant leaves, they also spread *Rhizoctonia solani* and ring rots. Colorado potato beetles live through the winter as adults in the soil [3], the biggest damages are done by Colorado potato beetles winter beetles shortly after they leave the hibernaculum, and before they lay eggs [4], if uncontrolled, the pest can cause serious yield losses [5,6,7].

Colorado potato beetles were first found in 1993 in the Xinjiang Uygur Autonomous Region of China. Since then, Colorado potato beetles spread from west to east along the oases in the northern area of the Tian Shan Mountains, until they reached Mori Kazak Autonomous County in 2003. By 2005, they were distributed in 38 counties (districts, cities) in seven locations (states, cities), including the Yili River Valley, Tacheng, and Altay. In 2009, Colorado potato beetle distribution expanded to eight prefectures and 35 counties and cities in Junggar Basin north of Tianshan Mountain, covering an area of about 260,000 km^2^ [8]. In 2014, Colorado potato beetles were found in the Sino-Russian border area. According to the “National Agricultural Plant Quarantine Pest Distribution Administrative Region Directory” (2019), Colorado potato beetles are distributed in three provinces; the distribution area is Xinjiang Uygur Autonomous Region > Heilongjiang Province > Jilin Province [9].

In the real environment, abnormal temperature fluctuations in spring and extreme weather sometimes occur [10]. From 1990 to 2000, the global incidence of extreme weather events was six times greater than events occurring in the mid-20th century [11,12,13]. Insects are ectotherms—their growth, development, and survival are affected by environmental factors, such as temperature, humidity, and precipitation. Temperature is one of the most important environmental factors affecting many aspects of the life cycles of insects [14]. Thermoregulation is critical for ectotherms as it allows them to maintain their body temperature close to an optimum for ecological performance, endotherms able to regulate their own body temperature, whereas ectotherms depend on the ambient temperature, so they depend on their behavior, physiology, morphology, etc., to regulate the body’s heat emissions or absorb heat from the outside environment to improve their body temperatures [15,16,17,18,19,20]. The fluctuations are larger than extreme high and low temperatures and can affect insect activities [21,22]. Low temperatures can reduce insect metabolism and affect development [23,24,25,26], overwintering survival [27], eclosion [28], death of the insect [29], although many insects have evolved mechanisms to cope with seasonal temperature changes [30]. However, abnormal seasonal temperature also will affect insect population dynamics [27].

Colorado potato beetles can migrate, allowing them to be widely distributed, and they have strong insecticide resistance, which makes them difficult to control [31]. The optimum development temperature of Colorado potato beetles is 25~30 °C [32], with the temperature range of 15 °C to 32 °C, within which each developmental stage of the Colorado potato beetle gradually becomes shorter. Colorado potato beetles can utilize genetic diversity, accumulated temperature to adapt to harsher environmental conditions [31,32], the supercooling point of diapause Colorado potato beetles are between −3.98 °C and 13.74 °C [33], and the lowest temperature of Colorado potato beetles development is between 6 °C and 11.5 °C [34]. Most studies of Colorado potato beetles at extreme temperatures, but less at minimum developmental temperatures. A cold spell in later spring is a weather phenomenon characterized by lower than normal temperatures [35]. An increase in soil temperature in the early stage makes Colorado potato beetles emerged to lay eggs. A cold spell (e.g., 8 °C) may occur; however, this temperature is realistic and closes to the lowest thermal threshold for developing Colorado potato beetles.

## 2. Materials and Methods

### 2.1. Insect Rearing

In this study, adults Colorado potato beetles (overwintered generation) were collected from a potato field (E 87.3831°, N 43.6132°, 1357 m) in Xishan Farm, a suburb of Urumqi, Xinjiang, without exposure to insecticides. Collected adults Colorado potato beetles were reared in an artificial climate chamber, the climate chamber (The climate chamber (RXM-168C-1 climate chamber, Ningbo Jiangnan Instrument Factory, Ningbo, China) were set at 27 ± 1 °C, 70% ± 5 RH, and a 16:8 h (L:D) photoperiod. Colorado potato beetles were fed daily with fresh potato leaves. The eggs from the third-generation adults were collected for the experiment.

The potato leaves used for Colorado potato beetles reared were from potted seedlings planted in 25 cm plastic pots under greenhouses. Plants were watered twice a week. When the potato plants were 25 cm tall, leaves were collected from the middle and upper parts of the potato plant. Collected potato leaves wrapped around the petioles of the potato leaves with wetted cotton for laboratory experiments.

### 2.2. Low Temperature Treatment of Colorado Potato Beetle Eggs

In the low temperature experiment, eggs were divided into five treatments; each group of 10–20 eggs was placed in a plastic culture dish. Plastic culture dishes were placed in a refrigerator for 0 d,1 d, 3 d, 5 d, 7 d, and 10 d at 8 °C (Stress temperature was measured with a Fuju thermometer (FOOJO, Tianjin, China), measuring accuracy: ±1 °C). This temperature was close to the lowest thermal threshold for developing Colorado potato beetles, and without incurring high mortality (The developmental zero temperatures for Colorado potato beetles’ whole stages is 8.1 °C [36]). 0 d: The eggs were placed in a climate chamber was set at 27 ± 1 °C, 70 ± 5% RH, and a 16:8 h (L:D) photoperiod. All treatments were replicated 5 times. Once removed from the refrigerator, eggs were placed in a climate chamber to hatch (The climate chamber was set at 27 ± 1 °C, 70 ± 5% RH, and a 16:8 h (L:D) photoperiod) with fresh potato leaves. Potato leaves were replaced daily, and the developmental periods and hatching rate were measured for the different treatments—each was inspected daily, living and dead larvae were counted.

### 2.3. The Emergence Survey of Adults Colorado Potato Beetles

The emergence data of adults Colorado potato beetles used in this study were obtained from Qapqal County Agricultural Extension Station, China. The survey was started before the Colorado potato beetles emerged (early April), each town randomly selected potato fields with the occurrence of Colorado potato beetles to be investigated. Five samples were selected, and installed the 1 × 1 m^2^ shroud in a field. The survey was conducted once daily. On each survey day, we recorded the number of emerged adults and removed them from the field [37].

### 2.4. Data Analyses

One-way ANOVA followed by LSD post-hoc comparisons were used to compare the significance of differences in developmental periods, hatching rate on different low temperature treatments (*p* < 0.05). Used SigmaPlot12.5 to draw the graphs of developmental periods, hatching rate, and emerged data of Colorado potato beetles.

The climate data used in this study were obtained from the Web site of China Meteorological Data Network (https://data.cma.cn, accessed on 15 October 2019). The daily climate data of Qapqal County in Xinjiang from 1993 to 2020 were used. The average temperature from 20 April to 5 May in each year was used as the study temperature (The Colorado potato beetle emerges from late April to early May [8]. For this study, we counted the number of days when the daily average temperature was below 12 °C, the number of times the daily average temperature was below 12 °C for 3 consecutive days, and the annual average temperature.)

## 3. Results

### 3.1. Effect of Different Durations of Low Temperature on Eggs

Different durations of low temperature exposure significantly affected the egg developmental duration of Colorado potato beetles (F = 139.335, df = 4, *p* < 0.05). Egg stage duration increased with the duration of low temperature exposure. The average egg stage duration (high to low) under different low temperature durations were 10 d > 7 d > 5 d > 3 d > 1 d > 0 d (Figure 1). The average egg period durations (low to high) were 4.35 d, 5.16 d, 7.04 d, 9.40 d, 10.76 d, 14.20 d.

### 3.2. Effect of Low Temperature Duration on Egg Hatching

The duration of low temperature exposure had a significant effect on egg hatching of the Colorado potato beetle (F = 22.281, df = 4, *p* < 0.05). The egg hatching rate under different low temperature durations were (high to low) 0 d > 1 d > 3 d > 5 d > 7 d > 10 d (Figure 2). The egg hatch rates were 91.60%, 86.69%, 79.85%, 75.53%, 60.46%, and 28.06%.

### 3.3. Colorado Potato Beetles Emerged in Qapqal County

Colorado potato beetles emerged in late April to early May, the earliest emerged date was 22 April, and the latest emerged date was 24 May (Figure 3).

The number of Colorado potato beetles emerged was positively correlated with the time of emerged in Qapqal County, the number of Colorado potato beetles emerged increased with the prolongation of emerged time, the number of emerged adults was about 5 per/m^2^ (Figure 4).

## 4. Discussion

Insects may be negatively impacted when temperatures are outside the range to which they are adapted. When the temperature is outside the tolerance range of an insect, and when exposed for long enough for damage to occur, mortality may increase, and activities can be inhibited [38,39,40,41,42]. Low temperature affects the development and survival of Colorado potato beetles. After 24 h of exposure to −5.8 °C, 50% of Colorado potato beetle eggs died [43]. In this study, Colorado potato beetle eggs examined at 8 °C and use the 27 °C for comparison, found that the negative effect of low temperature duration on Colorado potato beetle eggs, the duration of the egg stage increased, the percent of eggs hatching decreased as the duration of the low temperature stress duration increased, this result is consistent with the findings of Hiiesaar [43], and Colorado potato beetle eggs hatching rate was 89.51%, the development period was 4.35 d at 27 °C, this result is almost consistent with research under 27 °C [44].

Colorado potato beetles are highly adaptable and capable of rapid reproduction and dispersal [45,46], which makes them damage serious. Colorado potato beetle usually occurs for 1–2 generations in Xinjiang; when reached the appropriate temperature, Colorado potato beetles emerge to supplement nutrition and lay eggs (Colorado potato beetles live through the winter as adults in the soil) [4]. If Colorado potato beetle eggs lay in the cold spell in later spring, the egg hatching rate will reduce, retard eggs development, and delay subsequent population growth.

Temperature can affect the distribution of Colorado potato beetles [47], Qapqal County is located in the northern part of Xinjiang, belongs to the temperate continental climate, is a warm and dry regions—suitable for Colorado potato beetles growth [8,48]. Therefore, in this study, the emerged data of Colorado potato beetles and meteorological data from April to May were selected in Qapqal County to study the growth and development of Colorado potato beetles by short-term low temperature treatment. We found that Colorado potato beetles emerged from late April to early May in Qaqal County, it is similar to the observations of Li [49]. The temperature analysis of Qaqal County was found, during the time span of 1993 to 2020, the mean temperature in Qapqal County was 15.88 °C (Figure 5), the minimum temperature is 2.2 °C, this temperature deviates far below the normal temperature., and the mean daily temperature for these 18 years was below 12 °C, an average daily temperature below 12 °C for three consecutive days occurred in about seven years (Figure 6), consistent with the phenomenon of cold spells in late spring [36]. Tuerxn [4] found that when the temperature reached above 8 °C in April and the effective accumulated temperature reached 50 days, the adults began to emerge. If cold spells in late spring occurred before the Colorado potato beetles emerged, the number of emerged adults of the Colorado potato beetles would be decreased [50] (The number of Colorado potato beetles emerging was positively correlated with the emergence time in Qapqal County), which might reduce the damage of Colorado potato beetles.

## 5. Conclusions

Using Qapqal County as an example, our results demonstrate that low temperatures may occur in Qapqal County, and low temperatures have a negative effect on Colorado potato beetles, but this experiment only studied the hatching of Colorado potato beetle eggs, which was studied under laboratory conditions. In the field, the growth and development of the population were also affected by enemies, field planting environment, human factors, etc. The actual situation still needs further study and analysis.

## Figures and Tables

**Figure 1 insects-12-00609-f001:**
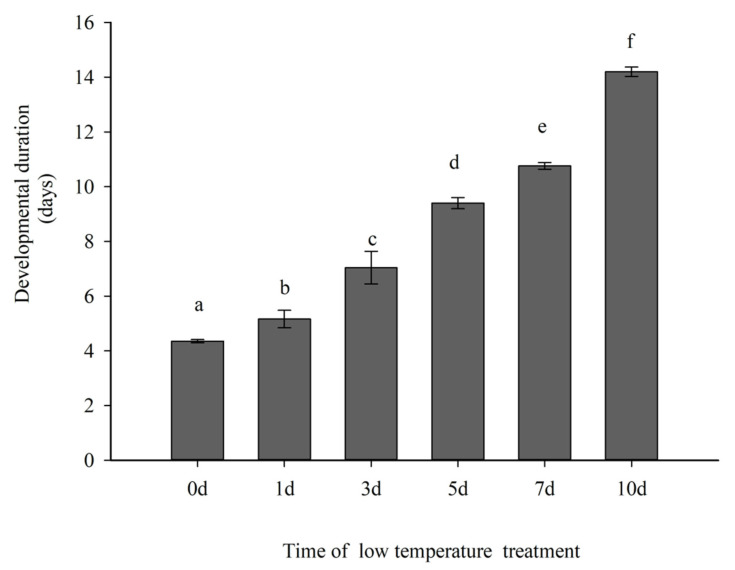
Development duration of Colorado potato beetle eggs under different durations at 8 °C. Mean ± SE development duration of Colorado potato beetle eggs following low temperature treatment durations. Different letters above the columns indicate statistically significant differences between durations (LSD test: *p* < 0.05).

**Figure 2 insects-12-00609-f002:**
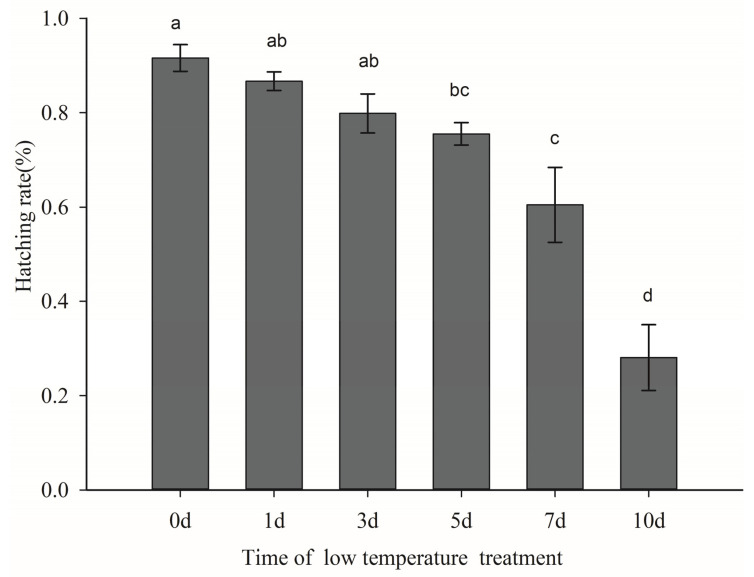
Hatching rate of Colorado potato beetle eggs exposed to different durations at 8 °C temperature. Mean ± SE hatch rate of Colorado potato beetle eggs under temperature treatments. Different letters above the columns indicate statistically significant differences in hatch time (LSD test: *p* < 0.05).

**Figure 3 insects-12-00609-f003:**
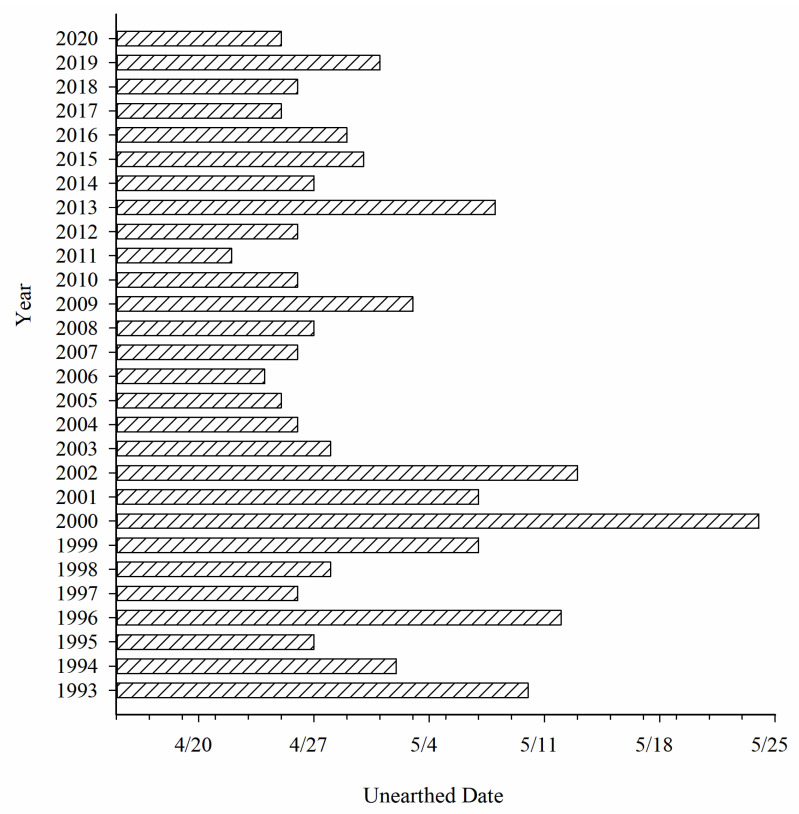
Emergence dates of Colorado potato beetles in Qapqal County (Date of first emergence).

**Figure 4 insects-12-00609-f004:**
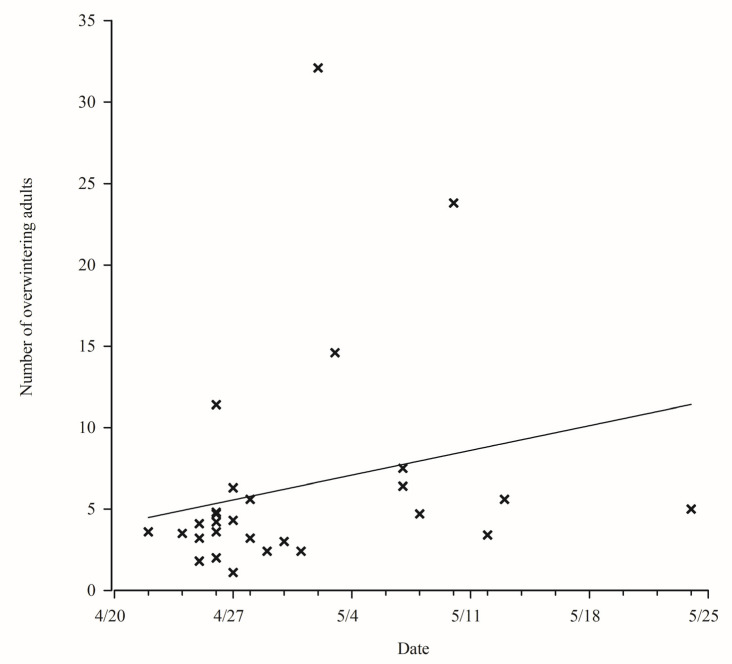
Cardinality map of Colorado potato beetles emerged from 1994 to 2010 and emerged over the winter in Qapqal County. The oblique line is the trend line. The figure “×” is a point of the number of Colorado potato beetles emerged.

**Figure 5 insects-12-00609-f005:**
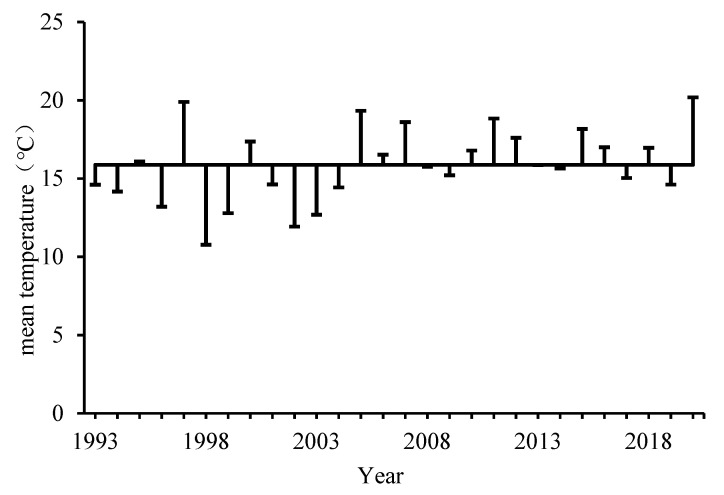
Average temperatures in Qapqal County. The horizontal line in the figure is the average year average temperature value in each selected time period. The annual average temperature is obtained by summing the average daily temperature of the selected date in each year and dividing it by the corresponding number of days. The annual average temperature is then obtained by summing the average daily temperature of each year and dividing it by the corresponding year.

**Figure 6 insects-12-00609-f006:**
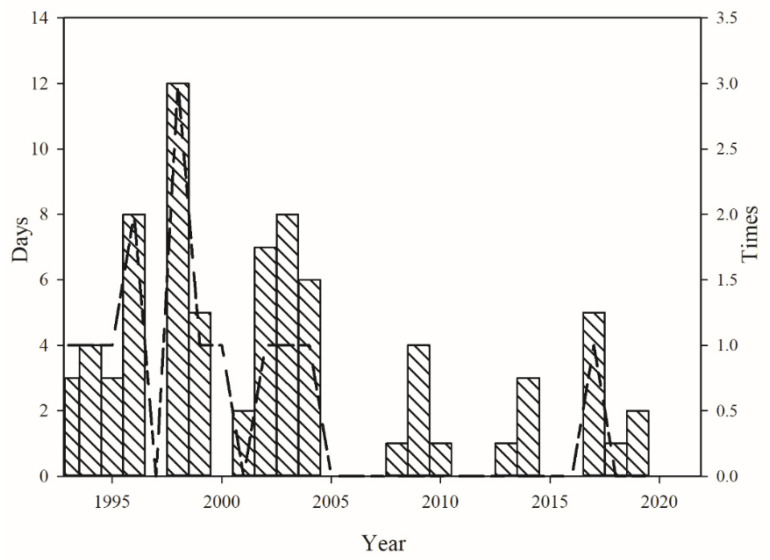
Temperatures below 12 °C in Qapqal County from 1993 to 2020. The histogram represents the number of days per year when the average daily temperature was lower than 12 °C. The line chart shows the number of times that the average daily temperature was below 12 °C for 3 consecutive days.

## Data Availability

Data are available upon request from the authors (3 July 2021).

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
