# Peer review of "Duration of Low Temperature Exposure Affects Egg Hatching of the Colorado Potato Beetle and Emergence of Overwintering Adults"

_insects, 2021, doi:10.3390/insects12070609_

Round 1

Reviewer 1 Report

This paper provides a straightforward study assessing the effect of cold exposure on the Colorado potato beetle. The general methods are sound, but I do have one concern with the analysis (see comments below pertaining to figure 2). The results are clear, and the discussion uses good science to interpret and discuss the results.

There are several run-on sentences that make reading the text a little difficult at times. The content is generally good, but should be broken up into more individual sentences at times (see some specific comments below for examples). There are also several spelling and other grammatical errors in the paper. I recommend a thorough read-through from an outside party.

Specific comments:

Simple summary – check to make sure you have periods and not commas. For example, in line 11, there should be a period after “China”, or “More” should not be capitalized.

Line 41: change “fed” to “feed”.

Lines 40-45: break this up into at least two sentences. This is a run on sentence.

Lines 61-62: this is not really a sentence. Are you saying that global warming causes frequent extreme climatic events and abnormal temperature fluctuations in the spring? Please clarify the sentence.

Lines 63-65: rewrite as “Because insects are ectotherms, there growth, development and survival are affected by environmental factors, such as temperature, humidity, and precipitation.

Lines 63-65: while the statement is not wrong, endotherms are also affected by all these factors. Maybe add something stating why ectotherms are more affected by these factors? Or why they are sensitive to these factors? That they do not internally regulate temperature like endotherms?

Line 71: what is meant by “overwintering”. I know what overwintering is generally, and diapause is one way organisms overwinter. This seems a bit vague here.

Lines 73-77: this is a run-on sentence. Break up for clarity.

Methods:

-how did you record the temperature of the refrigerator? Some type of thermometer/temperature logger?

Data analysis:

-I have concerns with the statistical analysis when looking at figure 2. How is 3d not significantly different than 0d when the SE does not overlap? Similarly, how is 7d significantly different than 5d when the SE do overlap? I certainly believe there are some significant differences, but should some bars be delineated as “ab” and “bc” with the overlap of several SE bars? If you provide the post-hoc/Fisher’s LSD test results in a table (possibly a supplemental table), this concern would be addressed immediately.

Lines 178-179: I generally agree, but duration of exposure is important. Perhaps add “when exposed for long enough for damage to occur”.

Line 194: change “dispera” to “dispersal”.

Line 204: change “affects” to “affect”.

Lines 227-228: change to “only indoors” or “was studied under laboratory conditions”.

Lines 236-241: while the plot contains interesting and useful data, the line chart is difficult to read. What about removing the hash lines from the histogram bars? This will likely make the line chart stick out more. And possibly make the line chart a dashed line so it sticks out even more from the histogram lines?

Author Response

Response to Reviewer 1 Comments

Point 1: There are several run-on sentences that make reading the text a little difficult at times. The content is generally good, but should be broken up into more individual sentences at times (see some specific comments below for examples). There are also several spelling and other grammatical errors in the paper. I recommend a thorough read-through from an outside party.(check to make sure you have periods and not commas.

For example, in line 11, there should be a period after “China”, or “More” should not be capitalized.

 Line 41: change “fed” to “feed”.

Lines 40-45: break this up into at least two sentences. This is a run on sentence.

Lines 61-62: this is not really a sentence. Are you saying that global warming causes frequent extreme climatic events and abnormal temperature fluctuations in the spring? Please clarify the sentence.

Lines 63-65rewrite as “Because insects are ectotherms, there growth, development and survival are affected by environmental factors, such as temperature, humidity, and precipitation.

Lines 178-179: I generally agree, but duration of exposure is important. Perhaps add “when exposed for long enough for damage to occur”.

Line 194: change “dispera” to “dispersal”.

Line 204: change “affects” to “affect”.

Lines 227-228: change to “only indoors” or “was studied under laboratory conditions

)

Response 1: spelling and rammatical errors had been modified.

Point 2: Lines 63-65: while the statement is not wrong, endotherms are also affected by all these factors. Maybe add something stating why ectotherms are more affected by these factors? Or why they are sensitive to these factors? That they do not internally regulate temperature like endotherms?

 Response 2: The explanation of why ectotherms are affected by these factors, why they do not internally regulate temperature like endotherms are added in the paper

Point 3: Line 71: what is meant by “overwintering”. I know what overwintering is generally, and diapause is one way organisms overwinter. This seems a bit vague here.

Response 3: This article would like to quote the way insects cope with adverse environment. Diapause is a way of insect overwintering, so this paper has been modified, citing the literature as insect overwintering, so diapause has been removed.

Point 4: Lines 73-77: this is a run-on sentence. Break up for clarity.

Response 4:It has already been modified.

Point 5: how did you record the temperature of the refrigerator? Some type of thermometer/temperature logger?

Response 5: Stress temperature was measured with a Fuju thermometer. measuring accuracy: ±1°C

Point 6: I have concerns with the statistical analysis when looking at figure 2. How is 3d not significantly different than 0d when the SE does not overlap? Similarly, how is 7d significantly different than 5d when the SE do overlap? I certainly believe there are some significant differences, but should some bars be delineated as “ab” and “bc” with the overlap of several SE bars? If you provide the post-hoc/Fisher’s LSD test results in a table (possibly a supplemental table), this concern would be addressed immediately.

Response 6:Significant labeling error, it has already been modified.

Table 1 ANOVE

ANOVE

Hacthing rate

Sum of Squares

df

Mean Square

F

Sig.

Between Groups

1.361

5

.272

22.281

.000

Within Groups

.293

24

.012

Total

1.654

29

Table 2 Multiple Comparisons

Multiple Comparisons

Dependdent Variable:  Hacthing rate

LSD

(I) Treatment times

(J) Treatment times

Mean Difference

(I-J)

SE

sig

95%Confidence interval

Lower Bound

Lower Bound

1d

3d

.0684000

.0698995

.338

-.075865

.212665

5d

.1116200

.0698995

.123

-.032645

.255885

7d

.2622800*

.0698995

.001

.118015

.406545

10d

.5862400*

.0698995

.000

.441975

.730505

CK

-.0491600

.0698995

.489

-.193425

.095105

3d

1d

-.0684000

.0698995

.338

-.212665

.075865

5d

.0432200

.0698995

.542

-.101045

.187485

7d

.1938800*

.0698995

.011

.049615

.338145

10d

.5178400*

.0698995

.000

.373575

.662105

CK

-.1175600

.0698995

.106

-.261825

.026705

5d

1d

-.1116200

.0698995

.123

-.255885

.032645

3d

-.0432200

.0698995

.542

-.187485

.101045

7d

.1506600*

.0698995

.041

.006395

.294925

10d

.4746200*

.0698995

.000

.330355

.618885

CK

-.1607800*

.0698995

.030

-.305045

-.016515

7d

1d

-.2622800*

.0698995

.001

-.406545

-.118015

3d

-.1938800*

.0698995

.011

-.338145

-.049615

5d

-.1506600*

.0698995

.041

-.294925

-.006395

10d

.3239600*

.0698995

.000

.179695

.468225

CK

-.3114400*

.0698995

.000

-.455705

-.167175

10d

1d

-.5862400*

.0698995

.000

-.730505

-.441975

3d

-.5178400*

.0698995

.000

-.662105

-.373575

5d

-.4746200*

.0698995

.000

-.618885

-.330355

7d

-.3239600*

.0698995

.000

-.468225

-.179695

CK

-.6354000*

.0698995

.000

-.779665

-.491135

CK

1d

.0491600

.0698995

.489

-.095105

.193425

3d

.1175600

.0698995

.106

-.026705

.261825

5d

.1607800*

.0698995

.030

.016515

.305045

7d

.3114400*

.0698995

.000

.167175

.455705

10d

.6354000*

.0698995

.000

.491135

.779665

*. The significance level of the mean difference is 0.05.

.

Point 7: Lines 236-241: while the plot contains interesting and useful data, the line chart is difficult to read. What about removing the hash lines from the histogram bars? This will likely make the line chart stick out more. And possibly make the line chart a dashed line so it sticks out even more from the histogram lines?

Response 7: The line chart has made the dashed line.

Reviewer 2 Report

Section 2.3 “The emergence survey of adults Colorado potato beetles” contains a description of the method for accounting for the emergence of adults, but these data are not described in the "Results" section, however are mentioned in the "Discussion". This analysis and graphs should be moved to the Results section. It is also necessary to give a more precise explanation of the scientific novelty of the study, since similar studies have been conducted since the 70s of the last century (possibly even earlier) and a lot of data have been accumulated on this issue.

Author Response

Response to Reviewer 2 Comments

Point 1: Section 2.3 “The emergence survey of adults Colorado potato beetles” contains a description of the method for accounting for the emergence of adults, but these data are not described in the "Results" section, however are mentioned in the "Discussion". This analysis and graphs should be moved to the Results section. It is also necessary to give a more precise explanation of the scientific novelty of the study, since similar studies have been conducted since the 70s of the last century (possibly even earlier) and a lot of data have been accumulated on this issue.

Response 1: The emergence survey of adults Colorado potato beetles data have been put into the results section and analyzed

Round 2

Reviewer 1 Report

General

While the paper is much improved, there are still some minor typos and grammatical errors in the manuscript. It appears that many of the brackets for the citations are lacking spaces from the preceding words. I point out some below, but the paper could use another read through. The content is good, so possibly a person not associated with the paper could help?

Specific

Lines 11-12: I believe the authors have placed commas where periods should be.

Line 44: delete space after period and add a space after the period (before “As Colorado”).

Lines 61-62: I do not follow this general statement. It reads as a random fact. Possibly expand or link to the proceeding sentence?

Line 67: while not wrong, this statement could be a little more specific as there are several thermoregulatory mechanisms (e.g., behavioral, physiological, and morphological). So maybe be more specific to ectotherms and just say “behavioral thermoregulation” as they are physiologically limited.

Line 95: remove citation #33 superscript.

Line 188: looking and figure 3 and the figure legend, it is unclear if this is the first date of beetle emergence. Please clarify in the legend.

Lines 251-257: the mean temperatures are interesting, but what about adding in error bars in the form of standard deviations to show the levels of variance?

Lines 257-258: is it necessary to include every year on the x-axis? This makes the axis very busy.

Author Response

Response to Reviewer 1 Comments

Point 1: Lines 11-12: I believe the authors have placed commas where periods should be.

Line 44: delete space after period and add a space after the period (before “As Colorado”).

Line 95: remove citation #33 superscript.

Response 1: They have already been modified.

Point 2: Lines 61-62: I do not follow this general statement. It reads as a random fact. Possibly expand or link to the proceeding sentence?

 Response 2: It has already been modified.

Point 3: Line 67 while not wrong, this statement could be a little more specific as there are several thermoregulatory mechanisms (e.g., behavioral, physiological, and morphological). So maybe be more specific to ectotherms and just say “behavioral thermoregulation” as they are physiologically limited.

Response 3: This statement have been changed to a more specific statement.

Point 4: Line 188: looking and figure 3 and the figure legend, it is unclear if this is the first date of beetle emergence. Please clarify in the legend.

Response 4:It has already been clarified in the legend..

Point 5: Lines 251-257: the mean temperatures are interesting, but what about adding in error bars in the form of standard deviations to show the levels of variance?

Response 5: It has already been modified.

Point 6: Lines 257-258: is it necessary to include every year on the x-axis? This makes the axis very busy.

Response 6: It has already been modified.